# Development and validation of self-monitoring auto-updating prognostic models of survival for hospitalized COVID-19 patients

Todd J. Levy [1,2], Kevin Coppa[3], Jinxuan Cang [1,2], Douglas P. Barnaby [1,4], Marc D. Paradis[5], Stuart L. Cohen[1,4], Alex Makhnevich [1,4], David van Klaveren [6,7], David M. Kent[7], Karina W. Davidson [1,4], Jamie S. Hirsch [1,3,4] & Theodoros P. Zanos [1,2,4] ✉

Clinical prognostic models can assist patient care decisions. However, their performance can drift over time and location, necessitating model monitoring and updating. Despite rapid and significant changes during the pandemic, prognostic models for COVID-19 patients do not currently account for these drifts. We develop a framework for continuously monitoring and updating prognostic models and apply it to predict 28-day survival in COVID-19 patients. We use demographic, laboratory, and clinical data from electronic health records of 34912 hospitalized COVID-19 patients from March 2020 until May 2022 and compare three modeling methods. Model calibration performance drift is immediately detected with minor fluctuations in discrimination. The overall calibration on the prospective validation cohort is significantly improved when comparing the dynamically updated models against their static counterparts. Our findings suggest that, using this framework, models remain accurate and well-calibrated across various waves, variants, race and sex and yield positive net-benefits.

Clinical prognostic models are used as tools for clinical decision making. Several well established models such as the sequential organ failure assessment score SOFA[1], CURB-65[2], the Apgar score[3,4], the Nottingham prognostic index[4,5], and the Manchester triage system[4,6] have been developed and are in common use in hospitals. These prognostic models are traditionally static[7]. However, model performance can drift and models can lose their predictive abilities over time[7–12]. Drifts in the predictive performance of models can appear as a reduction in overall accuracy[13] or miscalibration[8,14,15] and can be due to

changes in patient characteristics[14–17], as well as new treatments, changes in preventative care, or changes in treatment algorithms[14]. These reductions in predictive performance of models can render these models less useful[18] or even misleading[18,19], emphasizing the need to detect and correct these performance drifts.

Various approaches to compensate for model performance calibration drift have been proposed in the past. Newer or recalibrated versions of existing models have occasionally been developed to account for various patterns of invalidity[13], when model parameters are

[1]Institute of Health System Science, Feinstein Institutes for Medical Research, Northwell Health, Manhasset, NY 11030, USA. [2]Institute of Bioelectronic Medicine, Feinstein Institutes for Medical Research, Northwell Health, Manhasset, NY 11030, USA. [3]Clinical Digital Solutions, Northwell Health, New Hyde Park, NY 11042, USA. [4]Donald and Barbara Zucker School of Medicine at Hofstra/Northwell, Northwell Health, Hempstead, NY 11549, USA. [5]Northwell Holdings, Northwell Health, Manhasset, NY 11030, USA. [6]Department of Public Health, Erasmus MC University Medical Center, Rotterdam, Netherlands. [7]Predictive Analytics and Comparative Effectiveness Center, Tufts Medical Center, Boston, MA, USA. ✉e-mail: tzanos@northwell.edu

no longer optimal, due to changes in the case mix of patients, patient outcomes, and updates to the standard of care provided for hospitalized patients. For example, EuroSCORE was initially developed to predict mortality after heart surgery[20], and it was updated twelve years later[14]. Another clinical prognostic model, QRISK, which predicts the risk of a patient developing a heart attack or stroke within ten years is updated annually[15,21,22]. Dynamic updating approaches[7] update the model parameters at a specific time window[11,23,24] or when deemed necessary[10–13]. For the latter, the frequency of updating depends on the dynamics of the performance drift of such models, making the case for monitoring of prospective model performance. While rapid and continuous geographic validation of a newly developed model is challenging[18,25–27], temporal validation within one site or institution can be feasible, due to continuous recording of patient data as part of the standard of care.

With many well-established prognostic models of disease, model performance drift is a slow process and frequent monitoring or updating might not be required. The COVID-19 pandemic manifested itself as multiple geographically and temporally distinct waves that exhibited different characteristics such as different heterogeneous patient populations and different standard treatment methods[28]. Moreover, virus variants, changing vaccination rates, waning of vaccine protection, and emergence of novel COVID-19 treatments also contributed to a highly dynamic environment for patient characteristics and outcomes[29–32]. A commonly reported change was related to the significant decrease of case mortality rate of hospitalized patients in certain areas[33–35], as well as the number of intubated patients[35] and overall hospitalized patients between waves. Since the ongoing novel coronavirus pandemic is a situation in which many prognostic models were developed[19,36,37] on geographically distinct populations alongside rapid changes in patient populations and standards of care[38], it is unsurprising that a growing number of studies are reporting drifts in the performance of such models across different geographic regions[39–41] or different temporal windows[39,40]. While the need for model updating in COVID-19 is clear, there are no proposed self-monitoring and automatically updating prognostic models for COVID-19 patients.

Our main objective was to develop a survival calculator for COVID-19[42] that self-monitors and automatically updates whenever needed. We examined the need to recalibrate by monitoring discrimination and calibration from three modeling approaches over time. We compared model performance without updating to various model updating techniques[13,23,24] with each of the modeling approaches. We built a framework that can be applied both in traditional prediction model methods as well as machine learning-based methods and enables them to be easily monitored and updated accordingly. These techniques and ideas were tested using data from nearly 35,000 COVID-19 patients across the three major virus variants during the pandemic (alpha, delta, omicron) and three prognostic model architectures: our custom generalized linear model (GLM), logistic regression, and gradient boosted decision trees. The analysis was performed for a 28-day time horizon. Performance of all models with and without the self-monitoring and auto-updating capabilities were measured, and sensitivity and decision-curve analyses were conducted.

## Results
### Patient characteristics
A total of 38,078 electronic health records (EHRs) were considered in this study. Of these, 3166 were excluded because they were either transferred to a hospital outside of the health system and their outcomes were unknown, were started on invasive mechanical ventilation prior to admission, or had a do not resuscitate order placed outside of five days of death. All patients admitted after April 3, 2022 were also excluded to enforce the 28-day follow-up period. The remaining patients ($n = 34,912$, Table 1) were included in the development

($n = 7346$), retrospective ($n = 1889$), and prospective ($n = 25,677$) validation cohorts. The included patients (combined development and retrospective versus prospective) had a median age of 65 years with an interquartile range of [IQR 54-76] versus 67 years [IQR 54–79]), and (40% versus 47%) were female. The overall 28-day survival percentages were 77.0% for the combined development and retrospective versus 89.1% for the prospective cohorts.

### Development of survival prognostic models
To determine the predictors of survival, training data were collected from patients hospitalized in 11 of 12 included Northwell Health hospitals (Fig. 1). Three different model types—generalized linear model with the least absolute shrinkage and selection operator (LASSO) penalization (the Northwell COVID-19 Survival Calculator (NOCOS))[42], logistic regression (LR) with LASSO penalization[43], and extreme gradient boosted decision tree (XGBoost)[44]—were trained on patients from the development cohort for 28-day survival. The optimal predictors of survival were similar across each model. Age, serum blood urea nitrogen, lactate, and red cell distribution width were chosen as predictors, either in a linear or logarithmic scale, for all three models; albumin serum was chosen as a predictor for two of the three models; and respiratory rate and platelet count were each chosen once. Sometimes the linear and log scale of a single measurement were used as predictors within a model.

### Validation of survival prediction model
The retrospective validation included data collected from patients discharged from Long Island Jewish Hospital ($n = 1889$ [1470 survived past 28 days]) (Fig. 1). Based on this data, the 28-day NOCOS Calculator had a discriminative performance of the area under the receiver-operating-characteristic curve (AUROC) of 0.772 [95% CI 0.762, 0.782] (Fig. 2a) and the area under the precession-recall curve AUPR of 0.912 [0.906, 0.918] (Fig. 2a) and had a calibration performance of the integrated calibration index (ICI) of 0.047 [0.042, 0.054] (Fig. 2b), given the 22% mortality.

The prospective validation based upon data collected from patients discharged from all 12 Northwell hospitals ($n = 25,677$ [22,876 with 28-day survival]) (Fig. 1), with 28-day NOCOS Calculator discriminative performance AUROC 0.758 [0.755, 0.762] (Fig. 2a) and AUPR 0.945 [0.944, 0.947] (Fig. 2a). While the overall AUROC dropped slightly between the retrospective and prospective validation cohorts ($p = 0.005$ one-sided unpaired $t$-test) and the overall AUPR significantly improved ($p < 0.001$ one-sided unpaired $t$-test), the overall calibration performance degraded significantly from ICI 0.047 [0.042, 0.054] to ICI 0.119 [0.117, 0.120] ($p < 0.001$ one-sided unpaired $t$-test) (Fig. 2b, c).

### Updating the survival prediction model
Due to calibration degradation during the course of the COVID-19 pandemic, a 2000-patient sliding window was incremented at 500-patient intervals, and the models were updated when the ICI was greater than 0.03 (Fig. 3). The window size and updating threshold were selected based on hyperparameter optimization (Supplementary Fig. 6). All updating methods for the 28-day NOCOS produce similar AUROCs (Table 2). All updating methods were significantly better than the original model without updating with regard to ICI, with a range of ICI improvement from 0.098 to 0.105, ($p < 0.001$ one-sided unpaired $t$-test) (Fig. 3, Table 2). We selected logistic recalibration as our updating method for the 28-day NOCOS Calculator for generalization purposes, since it requires fewer patients for updating and fewer parameters to tune and also retains the same predictors[45]; it's also worth noting that intercept only recalibration is almost as good as full logistic recalibration. The overall results of this updating method are presented in Fig. 4.

We also performed similar updates and comparisons for 28-day survival logistic regression and XGBoost models (Figs. 2–4). All

**Table 1 | Demographic, clinical, and laboratory data of COVID-19 patients hospitalized at northwell health**

| | All included patients | Alive 28 days | Died 28 days | Missing no. (%) |
|---|---|---|---|---|
| *n* | 34,912 | 29,984 | 4928 | |
| Alpha (% of alpha patients) | 25,009 | 20,967 (83.8) | 4042 (16.2) | 0 (0.0) |
| Delta (% of delta patients) | 3475 | 3138 (90.3) | 337 (9.7) | 0 (0.0) |
| Omicron (% of omicron patients) | 6428 | 5879 (91.5) | 549 (8.5) | 0 (0.0) |
| Female (%) | 15,805 (45.3) | 13,817 (46.1) | 1988 (40.3) | 0 (0.0) |
| Male (%) | 19,107 (54.7) | 16,167 (53.9) | 2940 (59.7) | 0 (0.0) |
| Age, y (%) | | | | |
| 18–40 (%) | 3434 (9.8) | 3334 (11.1) | 100 (2.0) | 0 (0.0) |
| 41–60 (%) | 9618 (27.5) | 8902 (29.7) | 716 (14.5) | 0 (0.0) |
| 61–80 (%) | 14,746 (42.2) | 12,298 (41.0) | 2448 (49.7) | 0 (0.0) |
| 81–106 (%) | 7114 (20.4) | 5450 (18.2) | 1664 (33.8) | 0 (0.0) |
| Race (%) | | | | |
| Asian (%) | 2818 (8.1) | 2398 (8.0) | 420 (8.5) | 0 (0.0) |
| Black (%) | 6403 (18.3) | 5537 (18.5) | 866 (17.6) | 0 (0.0) |
| Declined (%) | 225 (0.6) | 196 (0.7) | 29 (0.6) | 0 (0.0) |
| Other (%) | 7821 (22.4) | 6838 (22.8) | 983 (19.9) | 0 (0.0) |
| Unknown (%) | 888 (2.5) | 764 (2.5) | 124 (2.5) | 0 (0.0) |
| White (%) | 16,757 (48.0) | 14,251 (47.5) | 2506 (50.9) | 0 (0.0) |
| Ethnicity (%) | | | | |
| Declined (%) | 157 (0.5) | 140 (0.4) | 7 (0.1) | 0 (0.0) |
| Hispanic or Latino (%) | 5878 (16.8) | 5148 (17.2) | 730 (14.8) | 0 (0.0) |
| Not Hispanic or Latino (%) | 27,516 (78.8) | 23,537 (78.5) | 3979 (80.7) | 0 (0.0) |
| Unknown (%) | 1361 (3.9) | 1149 (3.8) | 212 (4.3) | 0 (0.0) |
| English (%) | 29,657 (84.9) | 25,564 (85.3) | 4093 (83.1) | 0 (0.0) |
| Length of stay, days (median [IQR]) | 5.92 [3.15, 11.26] | 5.74 [3.08, 10.72] | 8.16 [3.96, 14.67] | 0 (0.0) |
| Vented (%) | 3713 (10.6) | 1542 (5.1) | 2171 (44.1) | 0 (0.0) |
| Last emergency department vital sign measurement (median [IQR]) | | | | |
| Systolic blood pressure, mmHg | 129.00 [115.00, 145.00] | 129.00 [115.00, 145.00] | 126.00 [111.00, 143.00] | 1526 (4.4) |
| Diastolic blood pressure, mmHg | 73.00 [65.00, 82.00] | 74.00 [65.00, 82.00] | 70.00 [61.00, 79.00] | 1526 (4.4) |
| Heart rate, beats per minute | 88.00 [77.00, 100.00] | 88.00 [77.00, 99.00] | 90.00 [78.00, 103.00] | 1509 (4.3) |
| Respiratory rate, breaths per minute | 20.00 [18.00, 22.00] | 19.00 [18.00, 21.00] | 20.00 [18.00, 24.00] | 1589 (4.6) |
| Temperature, Celsius | 37.10 [36.70, 37.70] | 37.10 [36.70, 37.70] | 37.10 [36.70, 37.90] | 1889 (5.4) |
| Oxygen saturation, % | 97.00 [95.00, 99.00] | 97.00 [95.00, 99.00] | 96.00 [94.00, 99.00] | 1594 (4.6) |
| Body mass index, kg/m$^2$ | 27.80 [24.30, 32.30] | 27.90 [24.40, 32.30] | 27.30 [23.60, 31.60] | 15,643 (44.8) |
| Height, cm | 167.64 [160.02, 175.26] | 167.64 [160.02, 175.26] | 167.64 [160.02, 175.26] | 10,765 (30.8) |
| Weight, kg | 79.40 [68.00, 93.00] | 79.40 [68.00, 93.40] | 77.10 [65.52, 90.70] | 12,327 (35.3) |
| Comorbidities, % | | | | |
| Coronary artery disease (%) | 2672 (7.7) | 2110 (7.0) | 562 (11.4) | 8598 (24.6) |
| Diabetes (%) | 6790 (19.4) | 5620 (18.7) | 1170 (23.7) | 8598 (24.6) |
| Hypertension (%) | 12,293 (35.2) | 10,237 (34.1) | 2056 (41.7) | 8598 (24.6) |
| Heart failure (%) | 1484 (4.3) | 1115 (3.7) | 369 (7.5) | 8598 (24.6) |
| Lung disease (%) | 3194 (9.1) | 2693 (9.0) | 501 (10.2) | 8598 (24.6) |
| Kidney disease (%) | 1757 (5.0) | 1378 (4.6) | 379 (7.7) | 8598 (24.6) |
| Last emergency department laboratory result (median [IQR]) | | | | |
| White blood cell count, K/µL | 7.34 [5.40, 10.07] | 7.24 [5.34, 9.86] | 8.05 [5.73, 11.56] | 1851 (5.3) |
| Absolute neutrophil, No., K/µL | 5.51 [3.80, 8.01] | 5.38 [3.73, 7.77] | 6.49 [4.32, 9.66] | 2648 (7.6) |
| Automated neutrophil, % | 76.70 [68.30, 83.40] | 75.90 [67.60, 82.60] | 81.30 [73.80, 87.00] | 2618 (7.5) |
| Automated lymphocyte, No., K/µL | 0.93 [0.62, 1.37] | 0.96 [0.65, 1.41] | 0.75 [0.50, 1.11] | 2644 (7.6) |
| Automated lymphocyte, % | 13.10 [8.10, 19.90] | 13.90 [8.70, 20.50] | 9.60 [5.70, 15.00] | 2617 (7.5) |
| Automated eosinophil, No., K/µL | 0.01 [0.00, 0.05] | 0.01 [0.00, 0.06] | 0.00 [0.00, 0.02] | 2695 (7.7) |
| Automated eosinophil, % | 0.10 [0.00, 0.80] | 0.10 [0.00, 0.90] | 0.00 [0.00, 0.20] | 2637 (7.6) |
| Automated monocyte, No., K/µL | 0.51 [0.34, 0.74] | 0.51 [0.34, 0.74] | 0.48 [0.30, 0.72] | 2644 (7.6) |
| Automated monocyte, % | 7.00 [4.90, 9.60] | 7.10 [5.00, 9.70] | 6.00 [3.90, 8.60] | 2617 (7.5) |
| Hemoglobin, g/dL | 12.90 [11.40, 14.20] | 13.00 [11.50, 14.30] | 12.40 [10.60, 14.00] | 1850 (5.3) |
| Red cell distribution width, % | 13.70 [12.80, 15.00] | 13.60 [12.80, 14.80] | 14.40 [13.30, 16.00] | 1876 (5.4) |

**Table 1 (continued) | Demographic, clinical, and laboratory data of COVID-19 patients hospitalized at northwell health**

| | All included patients | Alive 28 days | Died 28 days | Missing no. (%) |
|---|---|---|---|---|
| Automated platelet count, K/µL | 214.00 [163.00, 280.00] | 216.50 [166.00, 283.00] | 197.00 [145.00, 263.00] | 1888 (5.4) |
| Serum sodium, mmol/L | 137.00 [134.00, 139.00] | 137.00 [134.00, 139.00] | 136.00 [133.00, 140.00] | 1879 (5.4) |
| Serum potassium, mmol/L | 4.10 [3.70, 4.50] | 4.10 [3.70, 4.50] | 4.20 [3.80, 4.70] | 2076 (5.9) |
| Serum chloride, mmol/L | 100.00 [97.00, 104.00] | 100.00 [97.00, 104.00] | 100.00 [96.00, 104.00] | 1882 (5.4) |
| Serum carbon dioxide, mmol/L | 24.00 [21.00, 26.00] | 24.00 [22.00, 26.00] | 23.00 [20.00, 25.00] | 1872 (5.4) |
| Serum blood urea nitrogen, mg/dL | 18.00 [12.00, 29.00] | 17.00 [12.00, 26.00] | 28.85 [18.00, 48.00] | 1873 (5.4) |
| Serum creatinine, mg/dL | 1.02 [0.80, 1.44] | 1.00 [0.79, 1.34] | 1.36 [0.97, 2.23] | 1871 (5.4) |
| eGFR mL/min/1.73m$^2$ | 70.00 [44.00, 92.00] | 73.00 [48.00, 94.00] | 46.00 [25.00, 72.00] | 2168 (6.2) |
| Serum glucose, mg/dL | 122.00 [104.00, 160.00] | 120.00 [103.00, 155.00] | 136.00 [111.00, 188.00] | 1872 (5.4) |
| Serum albumin, g/dL | 3.50 [3.10, 3.90] | 3.60 [3.10, 3.90] | 3.20 [2.70, 3.60] | 2225 (6.4) |
| Total serum bilirubin, mg/dL | 0.50 [0.30, 0.70] | 0.50 [0.30, 0.70] | 0.50 [0.40, 0.80] | 2229 (6.4) |
| Serum alkaline phosphatase,U/L | 78.00 [61.00, 103.00] | 77.00 [61.00, 101.00] | 82.00 [63.00, 112.00] | 2267 (6.5) |
| Alanine aminotransferase (ALT/SGPT), U/L | 29.00 [18.00, 48.00] | 29.00 [18.00, 48.00] | 29.00 [18.00, 48.00] | 2330 (6.7) |
| Aspartate aminotransferase (AST/SGOT), U/L | 37.00 [25.00, 59.00] | 36.00 [24.00, 57.00] | 48.00 [31.00, 76.00] | 2335 (6.7) |
| Serum C-Reactive Protein, mg/L | 8.50 [4.00, 15.12] | 7.93 [3.70, 14.06] | 12.20 [6.58, 20.42] | 23730 (68.0) |
| Serum Lactate, mmol/L | 1.70 [1.30, 2.20] | 1.60 [1.20, 2.10] | 2.00 [1.50, 2.90] | 15506 (44.4) |

Age is binned in this table for presentation purposes, and the age variable is used as a numeric variable in the models.
*eGFR* estimated glomerular filtration rate using the CKD-EPI equation, *IQR* interquartile range.

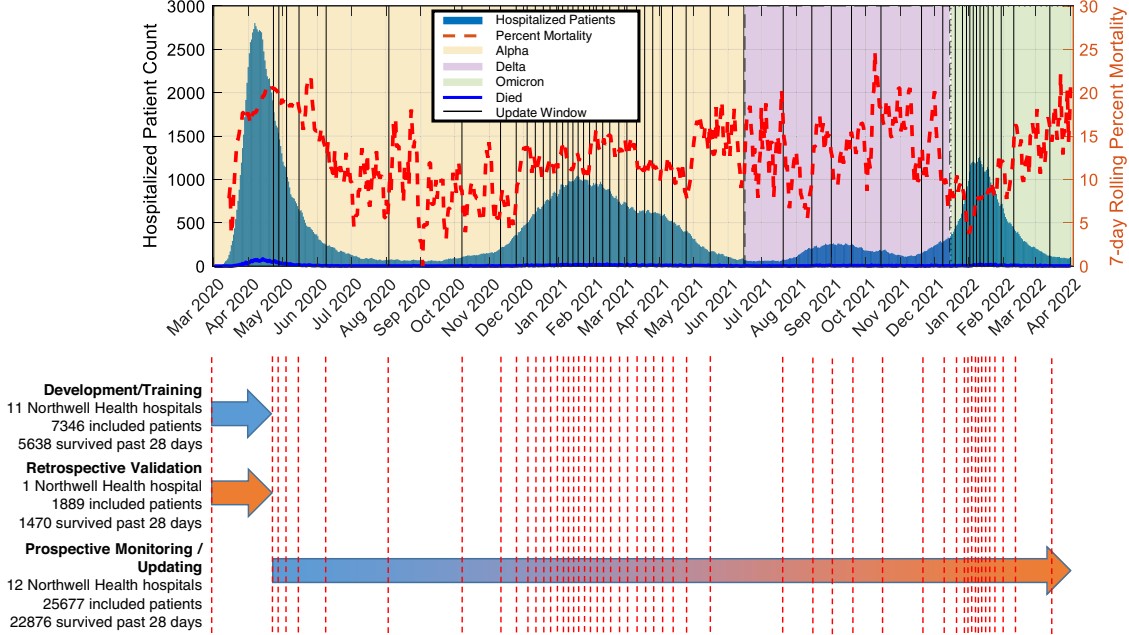

**Fig. 1 | Dataset overview and study design.** The data sets include training, retrospective validation, prospective monitoring/updating comprising 34,912 patients. The number of hospitalized patients across the course of the pandemic is plotted as blue bars, with the 7-day rolling average mortality plotted as a red dashed line. The three different dominant variants (alpha, delta and omicron) are represented as background colors). The vertical dashed red lines indicate the left edges of the 2000 patient sliding window that increments by 500 patients at a time. NOCOS Northwell COVID-19 Survival.

updating methods for LR and XGBoost yielded ICIs that were significantly better than not updating ($p < 0.001$ one-sided unpaired $t$-test) (Table 2). Neither logistic regression nor XGBoost, when updated, yielded a significantly lower ICI than the self-monitoring, auto-updating NOCOS. However, our analysis shows that updating is needed in all cases regardless of the model type whether it's a linear or nonlinear machine learning model. Since we favor updating methods with fewer parameters, and there were not drastic differences in performance across all of the different updating methods, we propose logistic recalibration for logistic regression and intercept only recalibration for XGBoost as the preferred updating methods.

### Changes in predictor importance
The dynamic Bayesian logistic regression is batch updated at each window position regardless of the current ICI estimate and provides a smoothly time-varying set of coefficients that can be compared to the selected update methods for the logistic regression model. The 28-day coefficients (Fig. 5) with logistic recalibration (Supplementary Fig. 4) are reasonable approximations to the dynamic Bayesian coefficients.

### Decision curve analysis
To better assess the clinical utility of the models, decision curve analysis was performed. The original yields a positive net benefit on the

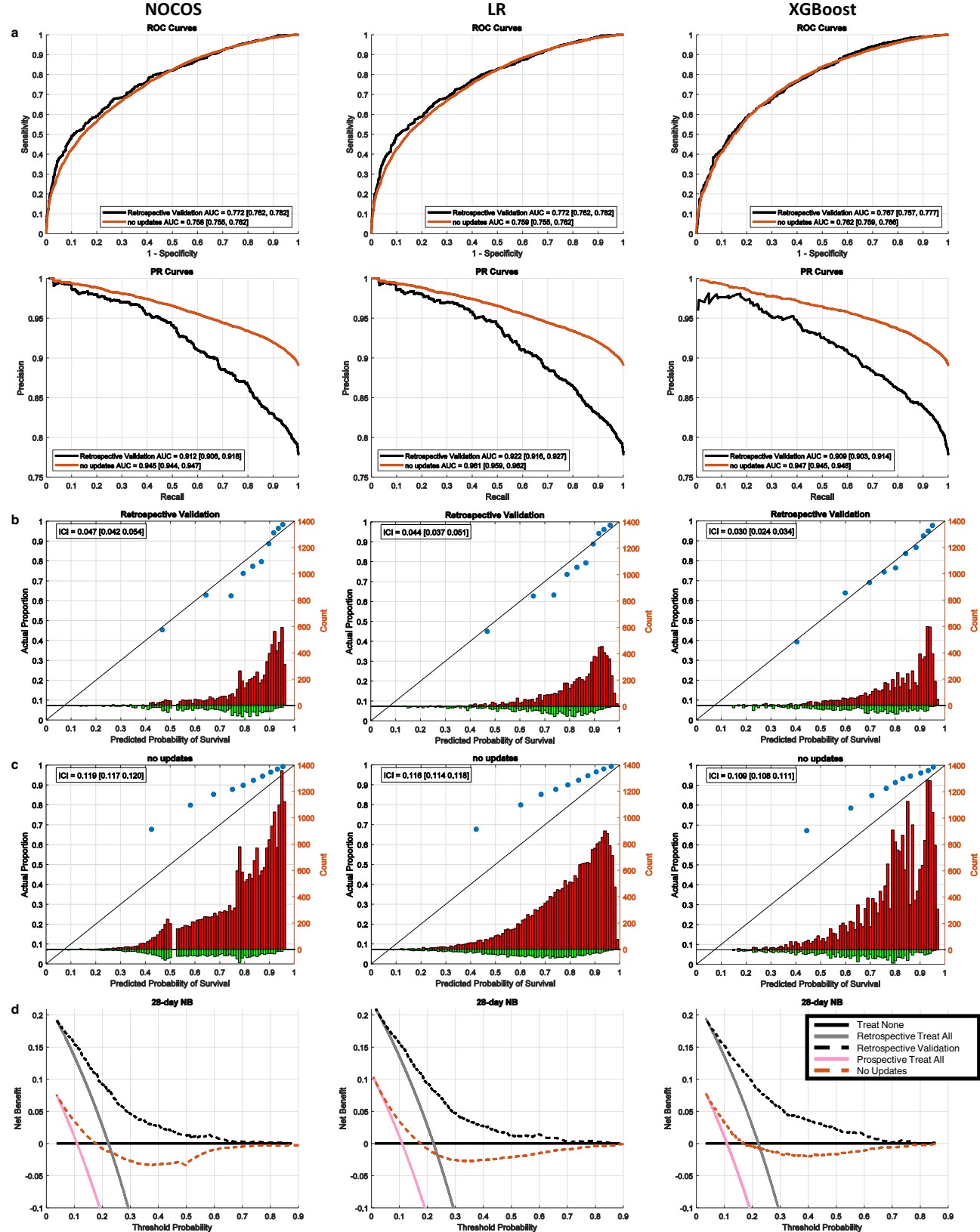

retrospective cohort across most decision thresholds (Fig. 2d). The original model without updating yields a negative net benefit over a wide range of preferences, showing that use of this model would worsen decision making, compared to the best treatment strategy overall. Worth reminding is that the probabilities used in our decision curve analysis refer to mortality, rather than survival. When we apply

the updating methods, the models tend to provide a positive net benefit. (Fig. 4c, Supplementary Fig. 3c).

## Sensitivity analysis
To examine whether the proposed framework remains accurate and well calibrated across different virus variants, sex and race/ethnicities,

**Fig. 2 | Retrospective and prospective validation of static 28-day survival models. a** ROC and PR curves with AUC and 95% CI for the retrospective (*n* = 1889) and prospective (*n* = 25,677; no updates) validation cohorts, **b** calibration plots for the retrospective validation cohort, **c** calibration plots for the prospective (no updates) validation cohort and **d** decision curves for the retrospective and prospective (no updates) cohorts based on the original NOCOS, logistic regression, and XGBoost models. The blue dots on the calibration plots show the actual proportion of outcomes averaged over deciles of the predicted probabilities. The red histograms show the counts of patients that survived past 28 days binned by the predicted probabilities. The green histograms show the counts of patients that died before 28 days binned by the predicted probabilities. The diagonal black lines indicate perfect calibration. The ICIs along with their 95% CIs are reported. ROC receiver operating characteristic, PR precision recall, AUC area under the ROC or PR curve, CI confidence interval, ICI integrated calibration index.

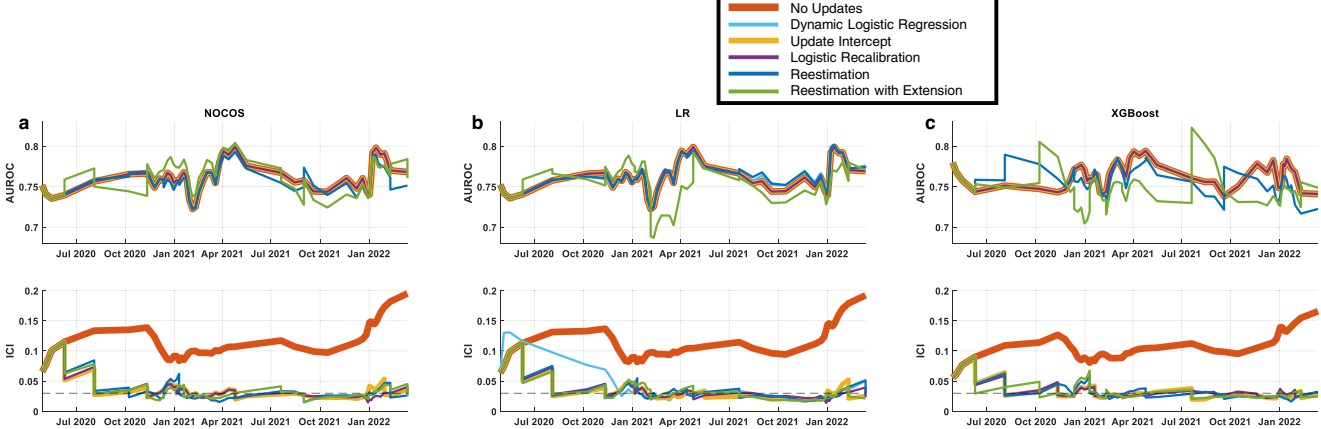

**Fig. 3 | Temporal progression of performance metrics across all 28-day survival models and updating procedures.** Discrimination (AUROC) and calibration (ICI) performance metrics in a 2000-patient sliding window with a step size of 500 patients for the original and dynamically updated 28-day **a** NOCOS, **b** logistic regression, and **c** XGBoost models. The updating methods are listed in the legend, and dynamic logistic regression is only applicable to the logistic regression model. Updates are performed when the ICI is greater than the threshold of 0.03. AUROC area under the receiver operating characteristic curve, ICI integrated calibration index, LR logistic regression.

## Table 2 | 28-day performance metrics for the prospective (*n* = 25,677) cohort

| AUROC [95% CI] | NOCOS | Logistic regression | XGBoost |
|---|---|---|---|
| No updates | 0.758 [0.755, 0.762] | 0.759 [0.755, 0.762] | 0.762 [0.759, 0.766] |
| Intercept only | 0.753 [0.750, 0.757] | 0.754 [0.751, 0.758] | **0.759 [0.756, 0.763]** |
| Logistic recal | **0.755 [0.752, 0.759]** | **0.756 [0.753, 0.760]** | 0.759 [0.756, 0.763] |
| Reestimation | 0.749 [0.745, 0.752] | 0.756 [0.753, 0.760] | 0.749 [0.745, 0.752] |
| Reest ext | 0.758 [0.754, 0.762] | 0.746 [0.742, 0.749] | 0.736 [0.732, 0.740] |
| Dynamic Bayes | N/A | 0.758 [0.754, 0.761] | N/A |

| AUPR [95% CI] | NOCOS | Logistic regression | XGBoost |
|---|---|---|---|
| No updates | 0.945 [0.944, 0.947] | 0.961 [0.959, 0.962] | 0.947 [0.945, 0.948] |
| Intercept only | 0.959 [0.957, 0.960] | 0.960 [0.958, 0.961] | **0.959 [0.958, 0.960]** |
| Logistic recal | **0.959 [0.958, 0.961]** | **0.960 [0.959, 0.961]** | 0.958 [0.957, 0.960] |
| Reestimation | 0.956 [0.955, 0.957] | 0.960 [0.959, 0.961] | 0.955 [0.954, 0.956] |
| Reest ext | 0.958 [0.957, 0.959] | 0.957 [0.956, 0.958] | 0.952 [0.951, 0.953] |
| Dynamic Bayes | N/A | 0.960 [0.959, 0.961] | N/A |

| ICI [95% CI] | NOCOS | Logistic regression | XGBoost |
|---|---|---|---|
| No updates | 0.119 [0.117 0.120] | 0.116 [0.114 0.118] | 0.109 [0.108 0.111] |
| Intercept only | 0.021 [0.020 0.022] | 0.017 [0.016 0.018] | **0.015 [0.014 0.016]** |
| Logistic recal | **0.016 [0.014 0.017]** | **0.015 [0.014 0.016]** | 0.016 [0.014 0.017] |
| Reestimation | 0.014 [0.013 0.016] | 0.018 [0.016 0.019] | 0.015 [0.014 0.016] |
| Reest ext | 0.019 [0.018 0.021] | 0.016 [0.015 0.018] | 0.017 [0.015 0.018] |
| Dynamic Bayes | N/A | 0.024 [0.023 0.026] | N/A |

Bold entries indicate the chosen update method for each model based on ICI.

*AUROC* area under the receiver-operating-characteristic curve, *AUPR* area under the precision-recall curve, *ICI* integrated calibration index, *NOCOS* Northwell COVID-19 Survival Calculator.

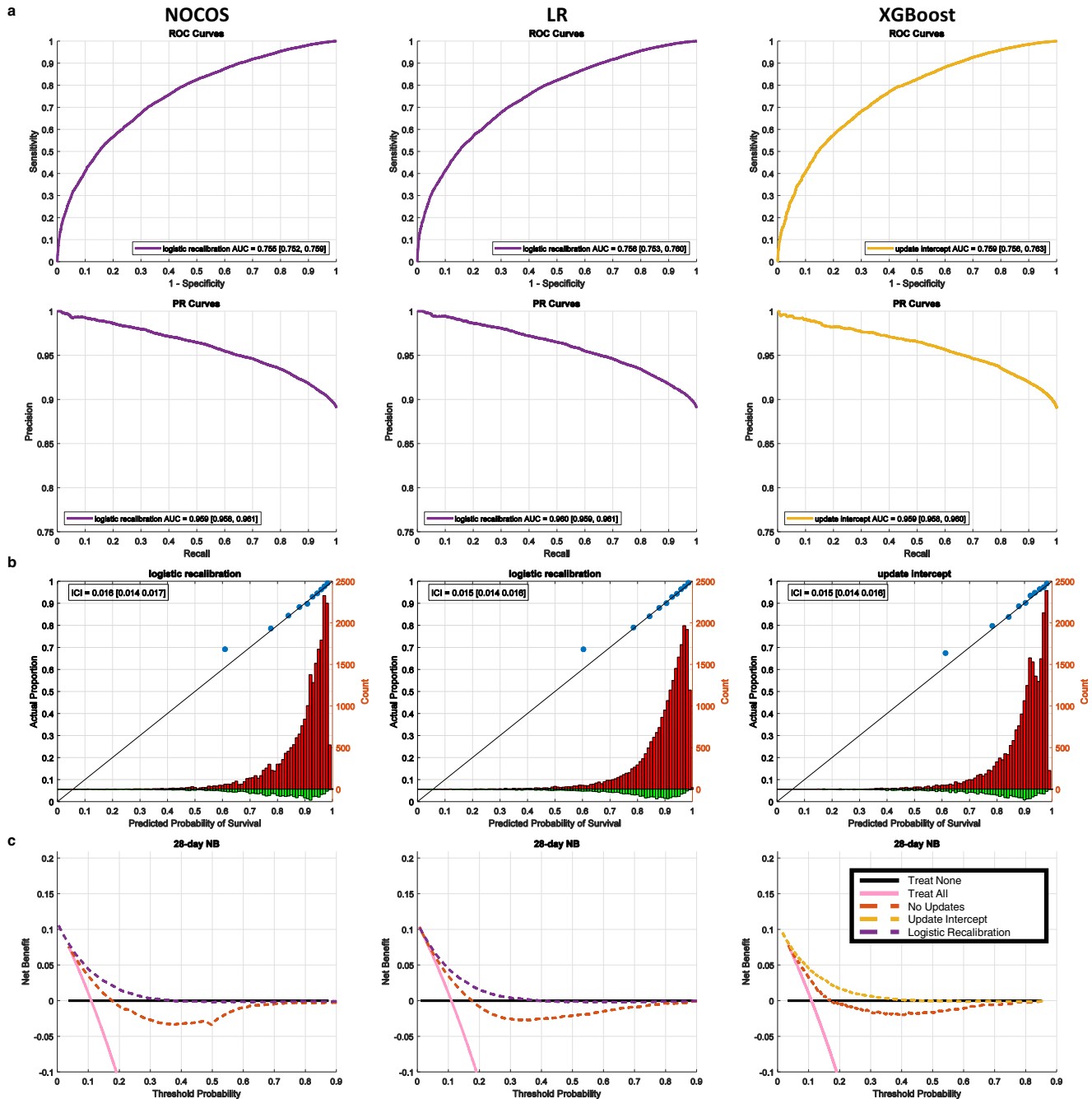

**Fig. 4 | Prospective validation of all 28-day self-monitoring, auto-updating models. a** ROC and PR curves with AUC and 95% CI for the prospective (*n* = 25,677) validation cohort, **b** calibration plots for the prospective validation cohort, and **c** decision curves for the prospective cohort based on NOCOS updated using logistic recalibration, logistic regression updated using logistic recalibration, and XGBoost updated using intercept only recalibration. The blue dots on the calibration plots show the actual proportion of outcomes averaged over deciles of the

predicted probabilities. The red histograms show the counts of patients that survived past 28 days binned by the predicted probabilities. The green histograms show the counts of patients that died before 28 days binned by the predicted probabilities. The diagonal black lines indicate perfect calibration. The ICIs along with their 95% CIs are reported. ROC receiver operating characteristic, PR precision recall, AUC area under the ROC or PR curve, CI confidence interval, ICI integrated calibration index.

we performed sensitivity analysis focusing on the performance characteristics of all models across these subgroups. Our results, shown in Fig. 6 and Supplementary Table 4 for the 28-day NOCOS model, as well as for all other models shown in Supplementary Fig. 7 and Supplementary Table 3, reveal small differences in performance metrics. The observed differences in ICI across all subgroups of our sensitivity analysis are not expected to affect the net benefit to the patient and showcase that the proposed approach works efficiently across virus variants, sex and races and ethnicities.

## Discussion

We developed a framework of self-monitoring, auto-updating prognostic models to predict the probability of 28-day survival for COVID-19 patients upon admission to the hospital. We applied this framework using three different model architectures: a custom GLM, logistic regression, and XGBoost. The same analysis was repeated for 7-day survival in the supplement. The initial onset of the first wave of the pandemic in New York City was a time when factors such as survival, standard treatment methods, and patient case-mix were under

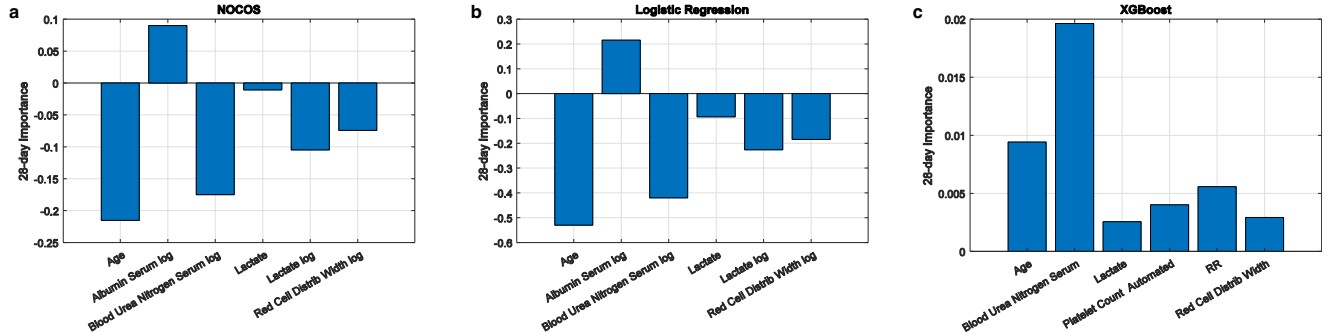

**Fig. 5 | 28-day model coefficient importance. a** NOCOS, **b** logistic regression, and **c** XGBoost model predictor importances are plotted. The importance of the NOCOS and logistic regression model predictors are the coefficients of the linear predictor scaled by the standard deviations of the predictors from the development cohort. The importance of the XGBoost model coefficients is the weighted average over the ensemble of trees of the difference in node risk between the parent and children nodes due to splitting at each predictor.

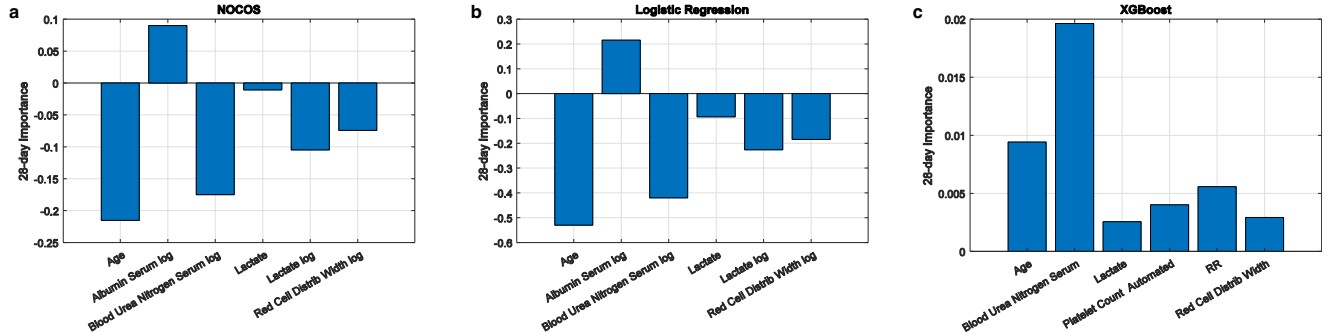

**Fig. 6 | Sensitivity analysis of the 28-day updating NOCOS model across variants, sex and race/ethnicity. a** ROC and PR curves with AUC and 95% CI for the prospective (*n* = 25677) validation cohort and **b** their corresponding calibration plots based on the 28-day NOCOS updated with logistic recalibration. The model was filtered by variant, sex, and race/ethnicity. The points on the calibration plot show the actual proportion of outcomes averaged over deciles of the predicted probabilities. The diagonal black lines indicate perfect calibration. The ICIs along with their 95% CIs are reported. ROC receiver operating characteristic, PR precision recall, AUC area under the ROC or PR curve, CI confidence interval; ICI integrated calibration index.

constant flux. These factors resulted in model performance drifts, showcased by the degradation of calibration performance from the retrospective to the prospective validation cohorts. In order to maintain calibration over time, a dynamic updating approach consisting of a 2000-patient sliding window with a 500-patient step was used to monitor the calibration and apply several model updating strategies if the miscalibration exceeded a threshold. The resulting models maintained good discrimination capabilities and calibration throughout the different waves of the pandemic, regardless of model architecture, always outperforming their initial versions. This is also the first study to our knowledge that performs temporal updating of COVID-19 prognostic survival models to correct for model performance calibration drift.

Prognostic models are frequently optimized for discriminative performance, but miscalibration can be harmful when clinical decisions are based on biased predictions[46]. The magnitude of the performance drift as well as the speed at which it manifests can have a potential impact on the perception of the severity of a patient. This can have an incremental effect on the treatment and, equally importantly, on the trust that clinicians have for these algorithms. While an error of 1–2% in a probability of survival estimate for a specific patient might not change any specific decision making, an average error of 9–10% across thousands of patients means that some patients will have an error a lot higher than 10% which can be misleading and result in confusion, delayed decision making, and loss of trust of the clinician to the algorithms. Our study emphasizes minimizing bias by reporting and maintaining calibration performance through model updates. Notably, despite calibration degradation, discriminative performance remained well-maintained, emphasizing the need for close monitoring of calibration characteristics (e.g., ICI) in addition to discrimination characteristics (e.g., AUC of ROC and PR curves).

We show that model updating is crucial, particularly in the setting of rapidly changing outcomes (e.g., survival), as in the course of the COVID-19 pandemic at our health system (Table 1) and elsewhere[47,48]. However, updating involves additional complexity, data gathering, and cost[7]. Ad hoc updated models like EuroSCORE are relatively inexpensive but are not responsive to calibration drift on a short timescale, and periodically updated models like QRISK are more expensive but react to calibration drift sooner. Dynamic models with continuous model surveillance have the benefit of increased responsiveness but also add complexity[24,49] to the validation steps since there is no standardized validation methodology[7–10]. We validated our models temporally by calculating the performance metrics in a sequence of sliding window positions, similar to the dynamic calibration curves used by Davis et al. albeit with a fixed window size and averaged over all predicted probabilities[12]. We also provided an overall validation of all models, by accumulating the predictions of the updated models and calculating the performance over all data, after initial development. Ultimately, standardized approaches to dynamic models that provide support in the context of healthcare systems will move us closer to learning health systems[50].

Our dynamic monitoring and updating method could be operationalized in a relatively straightforward manner. This deployment would feature a program that monitors the number of patients since deployment, and as soon as the number of patients reaches the required window size, will test the latest model calibration and, if needed, update the model. Creating that program is of minimal cost, and maintaining it can be rather trivial computationally, since these steps don't require any additional computational infrastructure. The minimum requirement is to maintain an actively updated database on a regular basis with a scheduled database query, employing in our use case specific filters for COVID-19 patients and notifications to inform clinicians of current performance and possible updates. As with every tool, it would be preferable to have a clinician, a developer. or ideally both, to review proper function of the pipeline, as well as the updates,

periodically. These processes can be run in a centralized location on a single workstation that interfaces with the electronic health records (EHR) of the institution it is deployed.

Another important factor to consider when updating models is the number of available samples. There is an inherent tradeoff between stationarity and sample size in the update cohort[51]. While small temporal windows can follow the dynamics of the nonstationary changes in the data, they might not have enough power in samples to enable certain modeling approaches, while larger windows that have adequate sample sizes might not be able to follow quickly changing dynamics[11]. In order to strike the balance for this tradeoff, our approach was to determine the size of the window based on a combination of formulas that estimate the proper sample size for LR models[52] and to perform a hyperparameter optimization that estimates the proper sample size for the custom GLM model (Supplementary Fig. 6). We ultimately recalibrated/retrained our existing model based on a 2000-patient sliding window of data rather than retraining the model on all of the data first.

We evaluated several updating methods: logistic recalibration with intercept only, full logistic recalibration with gain and intercept, retraining the model with fixed predictors, retraining the model with all candidate predictors[13], and dynamic Bayesian logistic regression[23,24]. In addition, other recent methods that smoothly update their parameters like the dynamic logistic state space model[53] may also be valuable within a dynamic updating framework. We selected the update method by finding the optimal integrated calibration index with minimal impact to discriminative performance. Most of the updating methods exhibited similar performance and all were superior to not updating, similar to other studies that employed model updating for other use cases[11]. In addition, our method included a lag equal to the follow-up period before updating the model to ensure that updating was done prospectively, unlike the approach by Schnellinger et al. that was inherently retrospective[11], which we found to be overly optimistic (see hyperparameter optimization and causal model design in Supplementary Fig. 6). Other methods in the literature for correcting model performance calibration drift include a closed test procedure[54], applying bootstrap resampling while evaluating the updating methods and scoring rules[10], and dynamically adjusting the window lengths[12]. These methods could also be implemented as a variation of our sliding window approach, but we found a single updating method with fixed window size sufficient without additional complexity. The closed testing procedure and retraining methods also require more data points than recalibration methods[11], and reestimation would not have been our optimal choice had fewer patients been initially available resulting in a reduced window size (see hyperparameter optimization in Supplementary Fig. 6).

Our results were also similar across model types, demonstrating robustness of the updating framework to the choice of model, whether that is a custom generalized linear model like NOCOS, a standard GLM like logistic regression, or a nonlinear ensemble model like XGBoost. It's interesting to note that XGBoost, a nonlinear machine learning model, did not yield significantly better results than a simple GLM or logistic regression model. This is likely because we constrained the models to only a few predictors or because the relationships in the data were mostly linear[55,56]. The results were also similar across the virus variant, sex, and race and ethnicity indicating that the updating methods are also robust to these variations in the data as well.

Updating a model, or more generally, making any change to a procedure, only makes sense if there is a clinical net benefit that results from the change. Since model performance calibration drift biases the model, we expected that there would be a net benefit from correcting the model calibration so that clinicians could make better decisions based on the model outcome. Decision curve analysis is a technique that graphs the net benefit of an intervention against a clinical preference[57]. All static models developed showed a negative net

benefit across a wide range of threshold values, indicating possible detriment if these models were used. When updated, the positive net benefit across a wide range of threshold preferences, indicates that these updates made the models generally useful, correcting for biases that can impact the net-benefit of these models.

Prognostic models for COVID-19 and other diseases are particularly useful clinically and operationally for health systems. For COVID-19 in particular, the NOCOS survival model was used extensively by triage teams (to better inform decisions about appropriate level of care and hospital discharge), clinical rounding teams (for risk assessment), primary medical and palliative care teams (to better guide shared medical decision making), and hospital operations teams (for load-balancing, resource, and personnel planning). Clinical decisions should not be based only on the predicted probability of survival, but also be based on a risk-benefit analysis that may be dependent upon current hospital resources[37]. Given these multiple uses with significant clinical and operational implications, ongoing monitoring, maintenance, and recalibration is necessary to maintain optimal performance.

The study population only included patients within the New York City metropolitan area. External, and more specifically geographic, validation of similar models has been limited and has demonstrated significant performance drifts[41,58]. While our model has been trained and validated using patients from the New York population, one of the most demographically and ethnically diverse areas in the world, it could still demonstrate performance drifts in different geographic locations. We believe, as demonstrated in this study, that our proposed framework can quickly correct for performance drifts, even those appearing immediately upon implementation.

The data were collected entirely from EHRs, which supported robust and rapid analysis of a large cohort of patients. However, we did not include data elements that would require manual chart review, such as symptom information or radiographs. Due to the retrospective study design, not all laboratory tests were completed on all patients, and the performance of these variables could not be adequately assessed.

Finally, deploying dynamic prediction models that self-monitor and auto-update can be technically challenging as they necessitate specific data pipelines and retraining scripts requiring maintenance and monitoring. One specific example may include the need for a data pipeline that is capable of dynamically updating the imputation models as well. Also, monitoring dashboards are essential, since engineering teams and stakeholders need to be alerted of model performance drifts and informed of any automatic updates, including specific details on changes occurring in updates. While these challenges can increase the technical burden of deploying these models, we believe that stable pipelines and data monitoring dashboards are necessary, not only for the case of dynamic models but for any deployed clinical predictive model.

This study demonstrates the importance of updating prognostic models in settings with rapidly changing clinical dynamics and proposes a self-monitoring, auto-updating survival model for COVID-19 patients. Biased models can result in potentially harmful biased clinical decisions. This is the first study to our knowledge that performs dynamic updating of COVID-19 prognostic survival models to correct for model performance calibration drift, a methodology that can be extended to other clinical prognostic models.

## Methods
### Data acquisition
Data were collected from the enterprise EHR (Sunrise Clinical Manager, Allscripts, Chicago, IL). Transfers from one in-system hospital to another were merged and considered one hospital visit. Data collected for the development and internal validation of the tool included patient demographic information, comorbidities, laboratory values,

and outcome (28-day survival and discharge). Our project utilized clinical data, obtained retrospectively, which was determined by the Northwell Health Institutional Review Board (IRB) to meet the requirements for review under exemption category 4) Secondary research uses of identifiable private information, (iii) The research involves only information collection and analysis involving the investigator's use of identifiable health information when that use is regulated under 45 CFR parts 160 and 164, subparts A and E, for the purposes of "health care operations" or "research" as those terms are defined at 45 CFR 164.501 or for "public health activities and purposes" as described under 45 CFR 164.512(b).

### Study design and setting
This study includes retrospective development and validation, and prospective validation of models to predict 28-day survival of patients hospitalized with COVID-19 between March 2, 2020, and April 3, 2022 (Table 1). The development cohort includes patients admitted to 11 acute care facilities in the Northwell Health system between March 2, 2020 (all date/times are at midnight), and April 23, 2020 ($n = 7346$). Long Island Jewish Hospital, the hospital that contained the most COVID-19 positive patients during the same time period, was left out of the development set, and was used for retrospective validation ($n = 1889$). We repeat this process, sequentially leaving one hospital out as the validation cohort, and we report the variation of retrospective performance metrics (Supplementary Table 5). The prospective validation cohort included patients admitted to all 12 acute care facilities in the Northwell Health system between April 23, 2020, and April 3, 2022 ($n = 25,677$). The final date of follow-up was May 1, 2022.

Patients were included in the development and retrospective and prospective validation datasets if they were adults (≥18 years old) admitted to the hospital with COVID-19 confirmed by a positive result from polymerase chain reaction testing of a nasopharyngeal sample. Clinical outcomes (i.e., discharge, mortality) were monitored until the final date of follow-up. Patients were excluded if they received invasive mechanical ventilation before admission, either before presentation to or during their stay in the emergency department. Another exclusion criterion is when a do not resuscitate order (DNR) was created up to five days prior to expiration, since these DNRs can potentially bias the outcome. Patients were also excluded if they were transferred to a hospital outside of the health system and their outcomes were unknown.

### Potential predictive variables
Potential predictive variables were included if they were available as discrete data points in the EHR for more than half of study patients at the time of admission. For variables with multiple values at admission, such as vital signs and labs, the last value before time of admission was used for analysis. This approach ensured that the results would contain data points routinely available upon admission. In the overview of our dataset (Table 1), continuous variables are presented as median and interquartile range, and categorical variables are expressed as the number of patients and percentage. Demographic variables included age, sex, race, ethnicity, and language preference as English or non-English. Vitals signs included systolic blood pressure, diastolic blood pressure, heart rate, respiratory rate, oxygen saturation, temperature, body mass index, height, and weight. Comorbidities included coronary artery disease, diabetes mellitus, hypertension, heart failure, lung disease, and kidney disease. Laboratory variables that have a missingness less than 50% included white blood cell count, absolute neutrophil count, automated lymphocyte count, automated eosinophil count, automated monocyte count, hemoglobin, red cell distribution width, automated platelet count, serum sodium, serum potassium, serum chloride, serum carbon dioxide, serum blood urea nitrogen, serum creatinine, estimated glomerular filtration rate, serum glucose,

serum albumin, serum bilirubin, serum alkaline phosphatase, alanine aminotransferase, aspartate aminotransferase, and serum lactate. C-reactive protein serum was also included even though it was more than 50% missing because other studies found it to be predictive of mortality[59,60]. The logarithms of numeric variables were used in addition to the linear scale. It is possible that both scales for a given measurement can be selected which can assist in modeling nonlinear effects. Categorical variables were dummy encoded by expanding the number of variables to include a binary representation of all but one of the categories where at most one variable at a time can be true.

## Imputation

Missing values were assumed to be missing at random and imputed using multiple imputations by chained equations (MICE)[61] using the R programming language, version 1.4.1717 (R Foundation for Statistical Computing). Some predictors like body mass index (BMI) and the natural logarithm of the laboratory values and vital signs were imputed using passive imputation to preserve the deterministic relationships between different missing variables in the imputed data. Outcomes were also included in the imputation models as recommended by Steyerberg in 7.4.3 of Clinical Prediction Models Second Edition since missing values were imputed using random draws[13]. Due to the nature of fully conditional specification and that the imputation models were trained on the development set, prospective outcomes are not necessary for imputing prospective data. Five imputed data sets were created using five random draws for each missing value. The imputation models were created on the development data, and new patients can be imputed using the same imputation models. Once the missing values have been imputed, the dynamic updating can proceed as if the values were known. We are effectively proposing method 6 from Hoogland et al.[62]. In a real-time implementation, to produce an individual risk prediction, a range of probabilities can be obtained from multiple imputation. This range along with the average estimated probability can be reported back so that an individual risk along with the impact of the missing values to this risk can be assessed by a clinician.

## Outcomes

Outcomes collected included death and discharge. The primary outcome was 28-day survival. Patients who were discharged alive within 28 days or were alive and in the hospital longer than 28 days had a positive survival outcome. Patients who died in less than 28 days had a negative survival outcome. The 28-day follow-up ensured that all outcomes were known for patients still in the hospital.

## Prediction model development

We previously developed the Northwell COVID-19 Survival Calculator (NOCOS); a generalized linear model that selected predictors from a pool of vitals, labs, and patient demographics using lasso regression[42,43]. NOCOS uses six variables to calculate a prognosis for in-hospital survival at the time of admission.

For this study, we monitored and retrained a NOCOS model for 28-day survival. The development cohort did not include one of the hospitals, Long Island Jewish Hospital and the cutoff date was modified to use patients that were admitted before April 23, 2020 rather than discharged before April 23, 2020, since the original NOCOS development set was inadvertently biased towards shorter duration patients. We also imputed the data using MICE with five imputations since the original NOCOS used mean imputation. Lastly, we made use of a more recent data set that includes patients admitted past our final date of follow-up, May 1, 2022 ($n = 34,912$, 29,984 survived 28 days) so that the 28-day follow-up period started on Apr 3, 2022.

The data were standardized by taking the z-score, which puts all measurements on the same scale. All analyses were performed in MATLAB 2020b (Mathworks, Inc., Natick, MA). The five imputed

development cohorts were combined via concatenation for model development[63] and weighted so as not to artificially increase the sample size, and the minority class (patients that died) was randomly oversampled with replacement[64,65] to correct for class imbalance. $L^1$-penalized linear regression followed by Bayes theorem was used to predict the survival of hospitalized patients with COVID-19 (Supplementary Methods). The class-conditional likelihood functions of the linear predictors for survival past 28 days and death before 28 days were estimated with Pareto tails and a Lévy alpha-stable distribution in the center using maximum likelihood estimation, and the priors were estimated as the fraction of patients that either survived or died. The posterior probability of survival past 28 days was evaluated using Bayes Theorem. Similar to logistic regression, linear regression followed by class-conditional likelihood estimation and Bayes theorem can also be formulated as a generalized linear model with a custom link function.

For comparison, we also developed an $L^1$-penalized logistic regression model and an extreme gradient boosted decision trees model for the 28-day outcome. The LR model did not require resampling, and the XGBoost model corrects for class imbalance via its cost function that incorporates the fraction of each outcome class. The XGBoost model can also be recalibrated just like the other models because it also predicts a probability. The hyperparameters for the LR models were selected to yield six predictors in the same way as NOCOS. Six predictors were selected for the XGBoost models by including all candidate predictors, ranking the predictors by their importance that is based on averaging changes in the node risk due to splits on every predictor[66], and then retraining the models using only the six most important predictors.

## Prediction model validation

The generalizability of the 28-day NOCOS calculator, as well as the LR and XGBoost models, were validated with the retrospective cohort and prospective cohort for each imputed data set. Predictive performance of the model was assessed via AUROC, AUPR, and ICI. AUPR is a measure that is well-suited for imbalanced data, and its values range from the sample prevalence (indicating random performance) to 1 (indicating perfect classification)[67]. ICI is the expected error between the actual and ideal calibration curves and approaches 0 when there is perfect calibration[68].

Ninety-five percent confidence intervals (95% CI) on the AUROCs were calculated using the method described by Hanley and McNeil[69], 95% CI for the AUPRs were calculated using the method described by Boyd, Eng, and Page[70], and the 95% CI for the ICIs were estimated using a bootstrapping approach with 200 replicates. All confidence intervals were determined over the union of the five imputed data sets. The confidence intervals for both area under the curves (AUCs) and ICIs were compared for significance using one-sided, two-sample t-tests, which are appropriate because AUCs are U-statistics, and the bootstrapped ICIs are observed to be approximately normally distributed.

## Prediction model updating

A sliding window was used for model surveillance and updating. The windowed patients were evaluated on the current model and the discrimination (AUROC and AUPR) and calibration (ICI) performance metrics were computed. This sweeps out performance versus time curves. In order to evaluate the overall performance, the predicted probabilities of the 500 most recent patients were accumulated over the course of the simulation and the performance metrics were recalculated over all of the accumulated data at the end of the simulation. If the ICI crossed a threshold, the model would be updated based on the current window of patients but not applied to new patients until after the follow-up since the patients' outcomes are not known at the time of admission. Hyperparameter optimization was performed by sweeping the window size, the ICI threshold, and the

number of imputations and partially rerunning the simulation (through November 15, 2021) to select the optimal values. Hyperparameters that are relatively insensitive to the performance would be expected to be robust with prospective data. The sliding window procedure was used for each updating method—no updating, intercept only logistic recalibration (updating the intercept of the linear predictor), full logistic recalibration (updating the intercept and applying a gain to the linear predictor), reestimation of the model parameters using the same predictors, and reestimation and selection of the model parameters using all candidate predictors[13]. The LR model also included a dynamic Bayesian logistic regression[23] updating method that was batch updated at each window position without a threshold.

In addition, a realizable model was compared with an acausal, unrealizable model. When causality is off in the simulation, patient outcomes are assumed to be known immediately, and the updated model is applied to new patients prior to the follow-up period. This results in an overly optimistic performance. We aimed to quantify the optimism bias based on this assumption for comparison, and the causal model used in all other experiments.

### Decision curve analysis

Decision curve analysis was performed by plotting the net benefit vs the preference. $NB(p_t) = TPR(p_t)*\varphi - w(p_t)*FPR(p_t)*(1 - \varphi) -$ $Net\ Harm$ where TPR is the true positive rate or sensitivity at the threshold probability $p_t$, FPR is the false positive rate at the threshold probability $p_t$, $\varphi$ is the estimated population event rate, and the weights $w = \frac{p_t}{1 - p_t}$ where $p_t$ is the threshold probability. The decision curve is compared against the treat-all (TPR = 1, FPR = 1) and treat-none (TPR = 0, FPR = 0) reference curves. In our decision curve analysis, the threshold probability refers to the mortality probability (in contrast to the survival probabilities of the models), to adhere to the commonly preferred way that decision curves are computed. There is no net harm from administering this test[71].

### Sensitivity analysis

Based on data from the New York State Department of Health[72], the alpha variant was dominant from the start of our data until about June 15, 2021. The delta variant was dominant between June 15, 2021 and December 15, 2021. Then the omicron variant was the dominant strain from December 15, 2021 through the end of our data. Race and ethnicity were combined so that White and Other Hispanics and Latinos were grouped with Hispanics and Latinos, but Black and Asian Hispanics and Latinos were grouped with Black and Asian respectively. The results of the primary analysis were indexed according to these groupings.

### Reporting summary

Further information on research design is available in the Nature Portfolio Reporting Summary linked to this article.

## Data availability

The tabular data that support the findings of this study are available in the synapse.org database - (https://doi.org/10.7303/syn39792658.1). The data being shared, acquired after a full waiver of authorization issued by the IRB, are not considered identifiable data as per the Safe Harbor method for deidentification articulated at 45 CFR 164.514(b)(2). Moreover, perturbation as per 45 CFR 164.514(b)(1) has been applied for additional reidentification risk mitigation.

## Code availability

Updating code can be downloaded from https://zenodo.org/badge/latestdoi/480857206 and a web calculator implementation is located at https://mlmd.org/nocos.

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

## Acknowledgements

We acknowledge and honor all our Northwell team members who consistently put themselves in harm's way during the COVID-19 pandemic. Their vital contribution to knowledge about COVID-19 and sacrifices on the behalf of patients made this possible. We would also like to acknowledge Challace Pahlevan-Ibrekic and Jackson Yeh for regulatory guidance and help with data de-identification and sharing. This work was supported by grants R24AG064191 from the National Institute on Aging, R01LM012836 from the U.S. National Library of Medicine, K23HL145114 from the National Heart, Lung, and Blood Institute (KWD), and ME-1606-35555 from the Patient-Centered Outcomes Research Institute (PCORI) Award (DvK, DMK, TPZ).

## Author contributions

T.J.L. designed the model and the computational framework, carried out the implementation, performed all data analysis, created all tables and figures, wrote the manuscript. K.C. collected and prepared the datasets. J.C. carried out the implementation of the online version of the model. D.P.B. provided critical feedback and assisted in writing the manuscript. M.D.P. provided critical feedback on aspects of model design. S.L.C. and A.M. provided critical feedback on the clinical interpretation and proofread the manuscript. D.vK. and D.M.K. provided critical feedback on statistical methods and model design, proofread the manuscript. K.W.D. provided critical feedback on clinical interpretation, assisted with planning and overall direction. J.S.H. assisted in the collection and preparation of the datasets, provided critical feedback on the model design and clinical interpretation and assisted in writing the manuscript. T.P.Z. conceived the study, designed the model and the computational framework, oversaw overall direction and planning, assisted in creating all tables and figures and wrote the manuscript.

## Competing interests

The authors declare no competing interests.
