## [Peer Review File · Nature Communications]

Reviewers' Comments:

Reviewer #1:

Remarks to the Author:

An interesting piece of work with a large data set. A well designed study incorporating comparable validation models. The potential benefits over static models were clearly described, although I am slightly uncertain how much small model performance drifts impact on clinical care, when models are used as an adjunct to clinical decision making only. Limitations are well addressed, although the practicalities and cost of applying and maintaining the dynamic updating system on a larger scale was not well discussed.

Reviewer #2:

Remarks to the Author:

The manuscript by Levy et al. implemented a few model updating strategies for predicting mortality risk in hospitalized COVID patients. They showed that the calibration was significantly improved for all updated models than the static counterparts. The concepts of dynamic model updating is not new but it seems the application to the COVID monitoring is novel.

Here are some detailed comments:

1. The data presentation can be streamlined by focusing on either 7-day or 28-day survival/mortality in the main manuscript and moving the other one to the supplement.
2. Line 150, please explain what "one-hot encoded" means.
3. Line 153, it seems appropriate to use multiple imputation to fill in the missing data when developing the prediction models. However, it is not clear how to use the same strategy in a prospective setting for both individual risk prediction and model updating. For example, it is not clear how to predict the mortality risk for a patient who misses some of the predictor values. Also, it is not clear how to include the patients with missing predictor information in the model updating.
4. Line 156, please explain what "passive imputation" means.
5. Line 168, please clarify the difference between the generalized linear model and logistic regression that were implemented in the manuscript.
6. Line 184, please explain how the "minority class was oversampled to correct for class imbalance"?
7. Line 185, it is not clear how Bayes theorem was used to predict survival. Is it specific to the generalized linear model? This is related to Comment #5. Please provide details how the generalized linear model is fit and used for prediction (may be in the supplement).
8. Line 205, the audience may not be familiar with AUPR and ICI. Please explain what they measure.
9. Line 216, please clarify how the "500 most recent patients were also stored for the overall performance".
10. Line 220, since the window size and threshold values should be prespecified for running the algorithm, it probably should not be called sensitivity analyses.
11. Line 224, in addition to the dynamic Bayesian logistic regression, there is also a recent publication related to this topic <https://doi.org/10.1111/biom.13593> that the authors may consider citing.
12. Line 291-292, the numbers do not seem to match with the figure. Please check.
13. Line 293-295, for the unpaired t-test for comparing performance, what is the analysis unit and corresponding sample size? Is it each sliding window?
14. Line 300, you mentioned threshold of 0.03 while it is 0.05 in Figure 3. Please clarify.
15. Line 321, please explain how to do calibration for XGBoost?
16. Line 324, my understanding is that dynamic Bayesian logistic regression (Ref 23) updated model coefficient when each new patient data becomes available. Please clarify how the model was implemented.

17. In Figure 2, panel A, please clarify the sample for which “no update” AUC and AUPR were calculated. The labels in the second row of figures should be AUPR instead of AUC?
18. In Figure 2, panel D, please explain what the two solid grey lines are.
19. In Figure 5, the first two models included lactate and log transformed lactate, which is hard to interpret. Please clarify what the candidate variables are in the model selection. Were the original and log transformed predictors included in the variable selection process simultaneously? Also, please include the distribution of lactate in Table 1.
20. In Figure 5, please explain how the coefficients were derived in XGBoost and why they are all positive.
21. In the title of Supplemental Figure 4, you mentioned “While the logistic regression coefficients do not change due to logistic recalibration, we plot the effect of changes in the calibration coefficients due to logistic recalibration, when combined with the logistic regression coefficients.” It is not clear why the logistic regression coefficients do not change. Please explain.
22. In Supplemental Figure 6, please explain what “causal and acausal models” are. Please clarify if all patients or just training data were included in these analyses. The error bars for different updating strategies overlapped quite bit, which makes it difficult to see. Please fix it.

REVIEWER COMMENTS

We wish to thank the reviewers for their thoughtful comments and suggestions. As you will see in this document, as well as the revised manuscript, the specific points raised by the reviewers have all been addressed; many clarifications have been added and some results, tables, p-values and figures have been modified/recalculated or moved to supplementary information. We believe all this additional work has significantly improved the paper and we hope the revisions are satisfactory to the reviewers and the editors. Our responses are highlighted in blue and the corresponding additions/changes to the revised manuscript are highlighted in red. All line numbers refer to the red-lined manuscript.

Reviewer #1 (Remarks to the Author):

An interesting piece of work with a large data set. A well designed study incorporating comparable validation models. The potential benefits over static models were clearly described, although I am slightly uncertain how much small model performance drifts impact on clinical care, when models are used as an adjunct to clinical decision making only. Limitations are well addressed, although the practicalities and cost of applying and maintaining the dynamic updating system on a larger scale was not well discussed.

- The reviewer is correct to point out that small performance drift might not necessarily have an impact on clinical care, especially since similar models like the one we proposed assist clinical decision making. However, the magnitude of the performance drift as well as the speed at which it manifests can have a potential impact on the perception of the severity of a patient. This can have an incremental effect on the treatment and, equally importantly, on the trust that clinicians have for these algorithms. While an error of 1-2% in a probability of survival estimate for a specific patient might not change any specific decision making, an average error of 9-10% across thousands of patients, as also displayed in our decision curve analysis, means that some patients will have an error a lot higher than 10% which can be misleading and result in confusion, delayed decision making, and loss of trust of the clinician to the algorithms. We believe that using our proposed approach can make sure that the errors are maintained at very low values, minimizing the risk of biased estimates as well as ensuring the maintenance of trust of all clinicians. **We have added these points in the discussion at paragraph lines 218-225 and lines 317-321.**
- Based on the reviewer comment, **we've now added a discussion paragraph on the practicalities and costs of applying this dynamic system on a larger scale at lines 246-**

257:

Our dynamic monitoring and updating method could be operationalized in a relatively straightforward manner. This deployment would feature a program that monitors the number of patients since deployment, and as soon as the number of patients reaches the required window size, will test the latest model calibration and, if needed, update the model. Creating that program is of minimal cost, and maintaining it can be rather trivial computationally, since these steps don't require any additional computational infrastructure. The minimum requirement is to maintain an actively updated database on a regular basis with a scheduled database query, employing in our use case specific filters for COVID-19 patients and notifications to inform clinicians of current performance and possible updates. As with every tool, it would preferable to have a clinician, a developer. or ideally both, to review proper function of the pipeline, as well as the updates, periodically. These processes can be run in a centralized location on a single workstation that interfaces with the EHR of the institution it is deployed.

Reviewer #2 (Remarks to the Author):

The manuscript by Levy et al. implemented a few model updating strategies for predicting mortality risk in hospitalized COVID patients. They showed that the calibration was significantly improved for all updated models than the static counterparts. The concepts of dynamic model updating is not new but it seems the application to the COVID monitoring is novel.

Here are some detailed comments:

1. The data presentation can be streamlined by focusing on either 7-day or 28-day survival/mortality in the main manuscript and moving the other one to the supplement.

Thank you for the suggestion to only include a single time horizon in the main manuscript and include the other time horizon in the supplement. You are absolutely correct that too many comparisons in the main manuscript will lead to confusion, and we have chosen to describe the 28-day models in the main manuscript and refer the supplement for the 7-day models. We made the following modifications:

1. Removed reference to 7 days in the abstract on line 30

2. Reworded lines 95-98 in the introduction to remove the reference to the 7-day models and refer to the supplement
3. Removed the reference to 7-days in the methods on lines 356, 381, 427-430, 437-438, 444, 454, 457, 470.
4. Removed the 7-day results on lines 108, 113-114, 121-126, 131, 139
5. Removed the references to the 7-day supplementary figures in the results on lines 122, 161, 164-165, 173, 178.
6. Removed references to 7 days in the discussion on line 203.
7. A sentence was added on line 205 to indicate that the same analysis was repeated for 7-day survival in the supplement.
8. The two 7-day columns were removed from Table 1 and a new Supplementary Table 1 was created that replaces the 28-day columns with the 7-day columns.
9. The 7-day survival statistics were removed from the left of Figure 1

2. Line 150, please explain what "one-hot encoded" means.

Thank you for pointing out our use of terminology. One-hot encoding is the machine learning equivalent of dummy variables in statistics. We expanded on the explanation on lines 407-409 to read:

categorical variables were dummy encoded by expanding the number of variables to include a binary representation of all but one of the categories where at most one variable at a time can be true.

3. Line 153, it seems appropriate to use multiple imputation to fill in the missing data when developing the prediction models. However, it is not clear how to use the same strategy in a prospective setting for both individual risk prediction and model updating. For example, it is not clear how to predict the mortality risk for a patient who misses some of the predictor values. Also, it is not clear how to include the patients with missing predictor information in the model updating.

You are correct that imputation in a prospective setting provides its own set of challenges. We chose an approach that always uses a fixed imputation model based on the development cohort. This is implemented on Line 107 in the imputation code (https://github.com/tlevy-nds/CPMupdate/blob/main/impute_COVID_reduced_20210302a.r). We can then impute any new patients using this same model. Once the missing values have been imputed, the dynamic updating can proceed as if the values were known. We are effectively proposing method 6 from Hoogland et al. (<https://onlinelibrary.wiley.com/doi/full/10.1002/sim.8682>).

In a real-time implementation, to produce an individual risk prediction, a range of probabilities can be obtained from multiple imputation. This range along with the average estimated probability can be reported back so that an individual risk along with the impact of the missing values to this risk can be assessed by a clinician.

We have inserted this description at lines 418-425 in the manuscript. We also included a sentence on the limitation that we used a fixed imputation model instead of a dynamically updated imputation model since that would require another updating framework on lines 338-339.

4. Line 156, please explain what “passive imputation” means.

Passive imputation is used when there is a deterministic relationship between two or more variables. For example, if two of the three variables in the equation $BMI = \text{weight}/\text{height}$ are unknown, passive imputation would impute the variables so that the relationship is maintained (See https://www.gerkovink.com/miceVignettes/Passive_Post_processing/Passive_imputation_post_processing.html). We expanded on the sentence on line 415 to clarify what passive imputation accomplishes.

5. Line 168, please clarify the difference between the generalized linear model and logistic regression that were implemented in the manuscript.

The original NOCOS model is described in reference 43 and estimates the probability of survival as follows:

$$P(Y = \text{survive} | X = x) = \frac{f_{X|Y=\text{survive}}(x)P(Y = \text{survive})}{\sum_{y \in \{\text{survive}, \text{die}\}} f_{X|Y=y}(x)P(Y = y)}$$

Lasso-penalized “linear” regression is used to select predictors. Let x be the linear predictor that is the inner product of the linear regression coefficients and the selected predictor values. The likelihood functions $f_{X|Y=\text{survive}}(x)$ and $f_{X|Y=\text{die}}(x)$ are estimated by fitting parametric distributions (Lévy alpha stable) to the class-conditional distributions of the linear predictor values for each of the two outcome classes; survive and die. The priors $P(Y = \text{survive})$ and $P(Y = \text{die})$ are calculated as the fraction of each type of outcome. One will also notice that the probability of survival calculation for the NOCOS GLM is a form of Bayes theorem.

Like the logistic regression model where the probability of survival is calculated using the inverse link function shown here,

$$P(Y = \textit{survive} | X = x) = g^{-1}(x) = \frac{1}{1 + e^{-x}}$$

the NOCOS model can theoretically be framed in the same way even if the link function cannot be determined symbolically. Technically speaking, NOCOS is a generalized linear model. While logistic regression is more widely known and established than GLMs in such applications, NOCOS is comparable in performance and faster computationally.

We have incorporated the above description for how to fit and predict with the NOCOS model into the supplementary methods and it is now referenced on line 452 in the main manuscript.

This description also answers reviewer question 7.

6. Line 184, please explain how the "minority class was oversampled to correct for class imbalance"?

Thank you for pointing out this important detail. The samples from the minority class (patients that died) were randomly sampled with replacement so that the number of patients that survived was equal to the number of patients that died. This is the random oversampling method mentioned in this article that lists several resampling strategies for imbalanced data (<https://www.ncbi.nlm.nih.gov/pmc/articles/PMC8641296/>). This oversampling method was only applied to the NOCOS model. It was not necessary to oversample the data for the logistic regression model, and the XGBoost model uses a cost function that incorporates the relative proportion of each class rather than oversampling the data. We modified the sentence at line 450, added a sentence at line 461 to incorporate these details, and added this reference in the manuscript.

7. Line 185, it is not clear how Bayes theorem was used to predict survival. Is it specific to the generalized linear model? This is related to Comment #5. Please provide details how the generalized linear model is fit and used for prediction (may be in the supplement).

See our answer to Question 5.

8. Line 205, the audience may not be familiar with AUPR and ICI. Please explain what they measure.

Thank you for pointing out the need for this clarification. We expect most readers will be familiar with AUROC, but AUPR and ICI aren't as common. The following description was added at lines 473-476:

AUPR is a measure that is well-suited for imbalanced data, and its values range from the sample prevalence (indicating random performance) to 1 (indicating perfect classification)⁴⁶ (<https://besjournals.onlinelibrary.wiley.com/doi/full/10.1111/2041-210X.13140>). ICI is the expected error between the actual and ideal calibration curves and approaches 0 when there is perfect calibration⁴⁷.

9. Line 216, please clarify how the “500 most recent patients were also stored for the overall performance”.

We appreciate this suggestion to help clarify our process.

The sliding window is 2000 patients long, and it moves in 500 patient steps. Predictions for all 2000 patients are evaluated and the ICI and AUROC are computed for the current window (See Figure 3). If required, the model is updated, and then the window is shifted. Our method measuring the “overall” performance was to accumulate each set of 500 new predictions and calculate the ROC and PR and calibration curves on the accumulated predictions at the end of the dynamic updating simulation (See Figure 4 A and B).

A couple new sentences were added at lines 485-489 to provide more details.

10. Line 220, since the window size and threshold values should be prespecified for running the algorithm, it probably should not be called sensitivity analyses.

We agree with the reviewer and replaced “sensitivity analysis” with “hyperparameter optimization” to avoid confusion with the sensitivity analysis that involves race/ethnicity, sex, and virus variant. We also changed the title of Supplementary Figure 6 to match this terminology. We added a couple sentences at lines 491-498 to elaborate on this point. We also updated this terminology at lines 151-152, 265, 280-281, 289.

We also added a short paragraph at lines 502-507 to explain the causality option:

Additionally, a realizable model was compared with an acausal, unrealizable model. When causality is off in the simulation, patient outcomes are assumed to be known immediately, and the updated model is applied to new patients prior to the follow-up period. This results in an overly optimistic performance. We aimed to quantify the optimism bias based on this assumption for comparison, and the causal model used in all other experiments.

11. Line 224, in addition to the dynamic Bayesian logistic regression, there is also a recent publication related to this topic <https://doi.org/10.1111/biom.13593> that the authors may consider citing.

Thank you for making us aware of this new technique. We cited it and added the following sentence at lines 272-274 in the discussion.

Additionally, other recent methods that smoothly update their parameters like the dynamic logistic state space model⁶⁰ may also be valuable within a dynamic updating framework.

12. Line 291-292, the numbers do not seem to match with the figure. Please check.

We thank the reviewer for finding this typo. We accidentally reported the AUROC and AUPR values from Figure 4 A instead of Figure 2 A. We updated the numbers in the manuscript at lines 141-142. We also recalculated the p-values (see the next Review Question).

13. Line 293-295, for the unpaired t-test for comparing performance, what is the analysis unit and corresponding sample size? Is it each sliding window?

This answer relates to the answer of Reviewer Question 9. The analysis unit is the "overall" set of accumulated predictions over the entire duration of the dynamic updating simulation. The corresponding sample size is the number of patients in the prospective cohort which is $N=25677$; the same number as given in Figure 1 and on line 139. This is compared against the retrospective cohort with a sample size of 1889.

We use the technique of Hanley and McNeil to obtain the confidence interval for the AUROC, we use the logit interval technique of Boyd et al. to obtain the confidence interval for the AUPR and transform it back to a linear space using the logit transform, and we use bootstrapping to obtain the confidence interval for the ICI which we verify to be close to normally distributed. We then solve for t in first bullet point of technical details section in https://cscu.cornell.edu/wp-content/uploads/73_ci.pdf and use the Welch-Satterthwaite equation to estimate the degrees of freedom, v , to evaluate the p-value for the unpaired t-test as $p = 1 - \text{tcdf}(t, v)$.

Because of the edits that resulted from Review Question 12, we recalculated the p-values and the p-value for the AUROC is now 0.005. The other p-values are still less than 0.001.

We updated the main manuscript to indicate the "overall" performance metrics on lines 142-144, and updated the p-value on line 143.

14. Line 300, you mentioned threshold of 0.03 while it is 0.05 in Figure 3. Please clarify.

Thank you for pointing out the error in our figure legend. We've corrected this in Figure 3 and in Supplementary Figure 2. The threshold should be 0.03.

15. Line 321, please explain how to do calibration for XGBoost?

Line 32 of <https://github.com/tlevy-nds/CPMupdate/blob/main/%40XGBoostBC/train.m> shows the implementation of our XGBoost classifier. It uses a ScoreTransform of logit and produces an output between 0 and 1. This probability can be transformed into a log odds ratio and recalibrated just like the linear predictors of the GLMs (<https://github.com/tlevy-nds/CPMupdate/blob/main/%40BinaryClassifier/recal.m>). We added the sentence "The XGBoost model can also be recalibrated just like the other models because it also predicts a probability" at lines 463-464.

16. Line 324, my understanding is that dynamic Bayesian logistic regression (Ref 23) updated model coefficient when each new patient data becomes available. Please clarify how the model was implemented.

You are correct that the dynamic Bayesian logistic regression model paper (McCormick et al.) describes a method for updating the model when the data for each new patient becomes available, but the paper also includes a section on batch updating (section 4.2.1) where it is assumed that each observation is i.i.d. within a batch. We implemented the batch updating method in our code (<https://github.com/tlevy-nds/CPMupdate/blob/main/%40LassoGlmBC/dynamicLR.m>) because it is more deployment friendly (updating every time a single patient record is available would add several implementation challenges) and is more comparable to the approach we are proposing.

We now explicitly state the batch update on lines 175 and 501.

17. In Figure 2, panel A, please clarify the sample for which "no update" AUC and AUPR were calculated. The labels in the second row of figures should be AUPR instead of AUC?

Thank you for pointing out this source of confusion. The figure caption discusses a comparison between the retrospective and prospective cohorts, and the figure legend labels the corresponding curves as retrospective and no updates. The no updates curve represents the prospective cohort when updating is turned off. To clarify this, we modified the figure caption to say "prospective (no updates)" for panel A and C. We also realized that we forgot to explicitly call out the prospective (no updates) curve in panel D, so we used the same terminology here as well. We made an identical change to the figure caption for Supplementary Figure 1 as well.

We also appreciate your point regarding the use of the terms AUROC, AUPR, and AUC. AUROC and AUPR are explicitly defined as the area under a ROC curve and the area under a PR curve respectively, and AUC is simply the area under the curve and can be the AUROC or AUPR depending upon the context. We now define AUC on line 534, and we changed the y-axis labels and figure captions from AUC to AUROC in Figure 3 and Supplementary Figure 2

because the context wasn't clear. We also redefine the context dependent AUC in the figure captions for all figures that include the term AUC.

18. In Figure 2, panel D, please explain what the two solid grey lines are.

You are correct to point out that decision curve analysis requires some extra explanation since it is far from being common knowledge. The gray curves are the reference curves (See Figure 1 in <https://diagnprognres.biomedcentral.com/articles/10.1186/s41512-019-0064-7>). Specifically, the dark grey line at $y=0$ is the treat-none reference curve, and each of the two light grey lines are the treat-all reference curves, and there are two because we are comparing the retrospective and prospective cohorts. The treat-all reference curves intersect the treat-none reference curves at the prevalence, so we can actually tell the mortality percentage of the retrospective and prospective cohorts from these plots.

We have changed the figure legends in Figure 2 D and Supplementary Figure 1 D to read "Treat None", "Retrospective Treat All", "Prospective Treat All" to clarify this point. The prospective treat all curve was changed to pink to differentiate it from the retrospective treat all curve.

We have also changed the figure legends in Figure 4 C and Supplementary Figure 3 C to read "Treat None" and "Treat All" since these figures are exclusively prospective.

The reference curves are already described in the methods, but the figure legends needed to match with our terminology.

19. In Figure 5, the first two models included lactate and log transformed lactate, which is hard to interpret. Please clarify what the candidate variables are in the model selection. Were the original and log transformed predictors included in the variable selection process simultaneously? Also, please include the distribution of lactate in Table 1.

Yes, the original and log-transformed predictors were included simultaneously. This was done to model simultaneously linear and possible nonlinear effects. We have added an explanation of this in the methods at line 405. And thank you for pointing out that we omitted lactate serum from Table 1 and the list of predictors on lines 397-403. We have now added an entry for Lactate in both places.

20. In Figure 5, please explain how the coefficients were derived in XGBoost and why they are all positive.

You are correct to point out that the wording of the figure 5 caption was incorrect. The generalized linear models contain coefficients, but since the XGBoost model is an ensemble of decision trees, it doesn't make sense to talk about coefficients. Instead we generalize this

to the concept of the importance of each predictor where the importance of the GLMs are the coefficients scaled by their respective standard deviations so that the inputs are standardized. The XGBoost importance is the weighted average over the ensemble of trees of the difference in node risk between the parent and children nodes due to splitting at each predictor

(<https://www.mathworks.com/help/stats/compactclassificationensemble.predictorimportance.html>). This is also why the importance of the predictors for XGBoost are not signed. **We reworded the figure caption to accurately describe what is being plotted.**

21. In the title of Supplementary Figure 4, you mentioned "While the logistic regression coefficients do not change due to logistic recalibration, we plot the effect of changes in the calibration coefficients due to logistic recalibration, when combined with the logistic regression coefficients." It is not clear why the logistic regression coefficients do not change. Please explain.

Using the notation of Steyerberg from Clinical Prediction Models Second Edition on page 402, logistic recalibration is given by the equation $lp_2(t) = \alpha_{new}(t) + \beta_{overall}(t) * lp_1$ where lp_1 is the linear predictor ($\beta_0 + \beta_1 * x_1 + \beta_2 * x_2 + \dots$), α_{new} is a calibration offset, and $\beta_{overall}$ is a calibration gain. In this updating method, lp_1 is static and $\alpha_{new}(t)$ and $\beta_{overall}(t)$ are updated to form $lp_2(t)$. Rather than plot β_{Age} , $\beta_{Albumin}$, etc. because they are unchanging across time, we plot $\alpha_{new}(t) + \beta_{overall}(t) * \beta_{Age}$, $\alpha_{new}(t) + \beta_{overall}(t) * \beta_{Albumin}$, etc.

We have updated the wording in the figure caption and added a brief description on lines 496-498.

22. In Supplementary Figure 6, please explain what "causal and acausal models" are. Please clarify if all patients or just training data were included in these analyses. The error bars for different updating strategies overlapped quite bit, which makes it difficult to see. Please fix it.

We updated the description of the hyperparameter optimizations at line 491 to include all four panels of Supplementary Figure 6. At lines 503-504, we also added the following sentence: "When causality is off in the simulation, patient outcomes are assumed to be known immediately, and the updated model is applied to new patients prior to the follow-up period".

We used all patients admitted before November 15, 2021 for the hyperparameter optimizations and also mention that at line 493. We can't use only the training data because these optimizations refer to the updating procedure, and it's preferable to use a subset of the data to perform hyperparameter optimization. The cutoff point of November 15, 2021 was chosen because that was the timepoint that we ran the hyperparameter optimization.

We also offset the traces in the figure so that the error bars won't completely overlap, and we say that we do this in the figure caption. We also added to the description of the causality of the models in the figure caption.

Lastly, we noticed that we were interchanging the terms gender and sex and corrected this to be consistent with Nature Communications' guidelines.

Reviewers' Comments:

Reviewer #2:

Remarks to the Author:

1. Line 415, you mentioned that outcomes were used in the imputation models. However, it is not clear how this imputation model can be used prospectively to impute missing predictors as the outcome is not known at the time of prediction.
2. The statistics section included only a small fraction of "statistics" that were used in the manuscript. I suggest you incorporate them into other sections.
3. Line 432-435, in addition to reference #42, a reference for the "GLM with LASSO selection method" should be provided.
4. Figure 1, please check the legend box in the top panel. It doesn't seem it is for this figure.
5. Figure 2, can you separate the retrospective and prospective validation results? The retrospective results can be moved to supplement. The figure in the main text should focus on the performance in prospective validation, which probably can be combined with Figure 4.
6. Given that the results from NOCOS and LR are almost identical (which makes sense since the two models are very similar), I wonder if it makes sense to move the NOCOS results to supplement to simplify the presentation, so that the main results on model updating can be further highlighted.

REVIEWER COMMENTS

We wish to thank the reviewers for their thoughtful comments and suggestions. Our responses are highlighted in blue and the corresponding additions/changes to the revised manuscript are highlighted in red. All line numbers in red and blue refer to the red-lined manuscript.

Reviewer #2 (Remarks to the Author):

1. Line 415, you mentioned that outcomes were used in the imputation models. However, it is not clear how this imputation model can be used prospectively to impute missing predictors as the outcome is not known at the time of prediction.

Thank you for pointing this out. Our imputation method deserves further clarification. When using multiple imputation, Steyerberg recommends including the outcome in the imputation model. We reproduce section 7.4.3 from Clinical Prediction Models Second Edition below. Also, when the outcome is missing, such as in the case of prospective data, the fully conditional specification that is used starts with a draw from the observed marginal distributions and iterates using a Gibbs sampler until convergence (<https://www.researchgate.net/publication/44203418> MICE Multivariate Imputation by Chained Equations in R, pages 6-7). Lastly, our imputation models were trained on the development set only, and applied to prospective data.

7.4.3 Imputation Models for SI and MI: X and y

For single imputation with the conditional mean (e.g., from a regression model), only the predictor variables should be used in the imputation model [594]. If the outcome y is also used in the imputation model, we exaggerate the strength of relations between predictors and outcome in the prediction model. In contrast, stochastic regression imputation should be performed with the outcome y . This is because a random element is added to the predicted values from the imputation model, similarly to an MI procedure (see Table 7.7).

We modified the relevant text (now at lines 512-516 in the revised manuscript) to specify exactly what we are referring to within the reference to make it more explicit for the reader:

Outcomes were also included in the imputation models as recommended by Steyerberg in 7.4.3 of Clinical Prediction Models Second Edition since missing values were imputed using random draws¹³. Due to the nature of fully conditional specification and that the imputation models were trained on the development set, prospective outcomes are not necessary for imputing prospective data.

2. The statistics section included only a small fraction of “statistics” that were used in the manuscript. I suggest you incorporate them into other sections.

We agree with this assessment. The statistics section expanded on information that was already present in the prediction model validation section. We moved the sentence describing the statistical comparison of the confidence intervals to the prediction model validation section as well and removed the statistics section.

3. Line 432-435, in addition to reference #42, a reference for the "GLM with LASSO selection method" should be provided.

Thank you for pointing this out. We added "Tibshirani R. Regression Shrinkage and Selection Via the Lasso. J R Stat Soc Series B Methodol. 1996;58(1):267-88." At line 534.

We also added references for each of the three methods on lines 141-143.

4. Figure 1, please check the legend box in the top panel. It doesn't seem it is for this figure.

Thank you for finding this mistake. We have now corrected the legend in Figure 1.

5. Figure 2, can you separate the retrospective and prospective validation results? The retrospective results can be moved to supplement. The figure in the main text should focus on the performance in prospective validation, which probably can be combined with Figure 4.

Thank you for this suggestion. However, we would still like to show the retrospective results in the main text because it establishes the motivation for why model updating is necessary. We first show that the retrospective results have good discrimination and calibration. In the same figure we show that the prospective results using the original model have degraded calibration so that the reader can directly comparable the two. Subsequently we show that this can be corrected by updating the model.

6. Given that the results from NOCOS and LR are almost identical (which makes sense since the two models are very similar), I wonder if it makes sense to move the NOCOS results to supplement to simplify the presentation, so that the main results on model updating can be further highlighted.

Thank you for the suggestion. However, we'd still prefer to keep both results: NOCOS and LR. Although they are close in performance, they do have slight differences in the way they are calculated, and computationally we found that NOCOS is more efficient, showcasing the usefulness of GLMs with LASSO. We still believe that the main results of model updating are highlighted across the three different modeling methods and the structure of the paper is maintained.